# Spawrious: A Benchmark for Fine Control of Spurious Correlation Biases

## Abstract

The problem of spurious correlations (SCs) arises when a classifier relies on non-predictive features that happen to be correlated with the labels in the training data. For example, a classifier may misclassify dog breeds based on the background of dog images. This happens when the backgrounds are correlated with other breeds in the training data, leading to misclassifications during test time. Previous SC benchmark datasets suffer from varying issues, e.g., over-saturation or only containing one-to-one (O2O) SCs, but no many-to-many (M2M) SCs arising between groups of spurious attributes and classes. In this paper, we present Spawrious-{O2O, M2M}-{Easy, Medium, Hard}, an image classification benchmark suite containing spurious correlations between classes and backgrounds. To create this dataset, we employ a text-to-image model to generate photo-realistic images and an image captioning model to filter out unsuitable ones. The resulting dataset is of high quality and contains approximately 152k images. Our experimental results demonstrate that state-of-the-art group robustness methods struggle with Spawrious, most notably on the Hard-splits with none of them getting over 73% accuracy on the hardest split using a ResNet50 pretrained on ImageNet. By examining model misclassifications, we detect reliances on spurious backgrounds, demonstrating that our dataset provides a significant challenge.

## 1 Introduction

One of the reasons we have not deployed self-driving cars and autonomous kitchen robots everywhere is their catastrophic behavior in out-of-distribution (OOD) settings that differ from the training distribution (D'Amour et al., 2020; Geirhos et al., 2020). To make models more robust to unseen test distributions, mitigating a classifier's reliance on spurious, non-causal features that are not essential to the true label has attracted lots of research interest (Sagawa et al., 2019a; Arjovsky et al., 2019; Kaddour et al., 2022b; Izmailov et al., 2022). For example, classifiers trained on ImageNet (Deng et al., 2009) have been shown to rely on backgrounds (Xiao et al., 2020; Singla & Feizi, 2022; Neuhaus et al., 2022), which are spuriously correlated with class labels but, by definition, not predictive of them.

Recent work has focused substantially on developing new methods for addressing the spurious correlations (SCs) problem (Kaddour et al., 2022b), yet, studying and addressing the limitations of existing benchmarks remains underexplored. For example, the *Waterbirds* (Sagawa et al., 2019a), and *CelebA hair color* (Liu et al., 2015) benchmarks remain among the most used benchmarks for the SC problem; yet, GroupDRO (Sagawa et al., 2019a) achieves 90.5% worst-group accuracy using group adjusted data with a ResNet50 pretrained on ImageNet.

Another limitation of existing benchmarks is their sole focus on overly simplistic one-to-one (O2O) spurious correlations, where one spurious attribute correlates with one label. However, in reality, we often face *many-to-many* (M2M) spurious correlations across groups of classes and backgrounds, which we formally introduce in this work. Imagine that during summer, we collect training data of two groups of two animal species (classes) from two groups of locations, e.g., a tundra and a forest in eastern Russia and a lake and mountain in western Russia. Each animal group correlates with a background group. In the upcoming winter, while looking for food, each group migrates, one going east and one going west, such that the animal groups

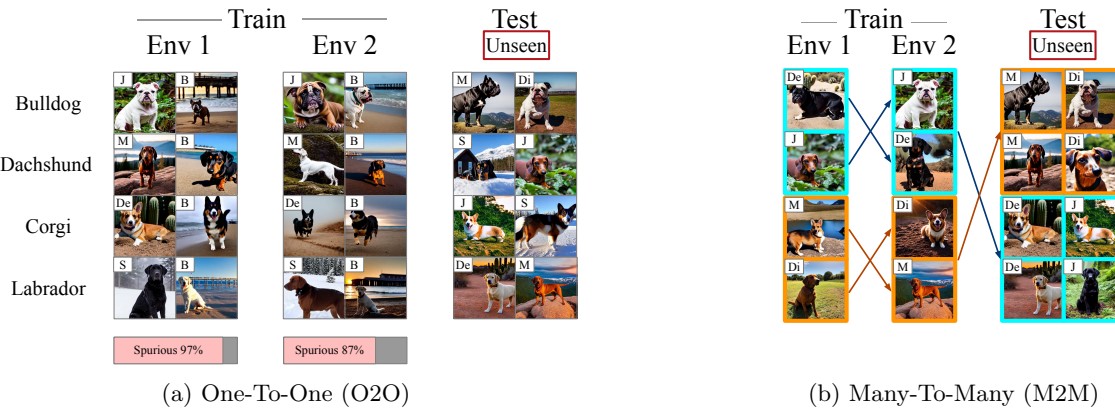

(a) One-To-One (O2O)                    (b) Many-To-Many (M2M)

Figure 1: **Spawrious Challenges:** Letters on the images denote the background, and the bottom bar in Figure 1a indicates each class's proportion of the spurious background. In the O2O challenge, each class associates with a background during training, while the test data contains unseen combinations of class-background pairs. In the M2M challenge, a group of classes correlates with a group of backgrounds during training, but this correlation is reversed in the test data.

have now exchanged locations. The spurious correlation has now been reversed in a way that cannot be matched from one class to one location.

While some benchmarks include multiple training environments with varying correlations (Koh et al., 2021), they do not test classification performance on reversed correlations during test time. Such M2M-SCs are *not* an aggregation of O2O-SCs and cannot be expressed or decomposed in the form of the latter; they contain qualitatively different spurious structures, as shown in Figure 2. To our knowledge, this work is the first to conceptualize and instantiate M2M-SCs in image classification problems.

**Contributions** We introduce *Spawrious*-{O2O, M2M}-{Easy, Medium, Hard}, a suite of image classification datasets with O2O and M2M spurious correlations and three difficulty levels each. Recent work (Wiles et al., 2022; Lynch et al., 2022; Vendrow et al., 2023) has demonstrated a proof-of-concept to effectively discover spurious correlation failure cases in classifiers by leveraging off-the-shelf, large-scale, image-to-text models trained on vast amounts of data. Here, we take this view to the extreme and generate a novel benchmark with $152,064$ images of resolution $224 \times 224$, specifically targeted at the probing of classifiers' reliance on spurious correlations.

Our experimental results demonstrate that state-of-the-art methods struggle with Spawrious, most notably on the *Hard*-splits with $< 73\%$ accuracy using ResNet50 pretrained on ImageNet. We probe a model's misclassifications and find further evidence for its reliance on spurious features. We also experiment with different model architectures, finding that while larger architectures can sometimes improve performance, the gains are inconsistent across methods, further raising the need for driving future research.

## 2  Existing Benchmarks

We summarize the differences between Spawrious and related benchmarks in Table 1. DomainBed (Gulrajani & Lopez-Paz, 2021) is a benchmark suite consisting of seven previously published datasets focused on domain generalization (DG), not on spurious correlations (excluding CMNIST, which we discuss separately). After careful hyper-parameter tuning, the authors find that ERM, not specifically designed for DG settings, as well as DG-specific methods, perform all about the same on average. They conjecture that these datasets may comprise an ill-posed challenge. For example, they raise the question of whether DG from a photo-realistic training environment to a cartoon test environment is even possible. In contrast, we follow the same rigorous hyper-parameter tuning procedure by (Gulrajani & Lopez-Paz, 2021) and observe stark differences among

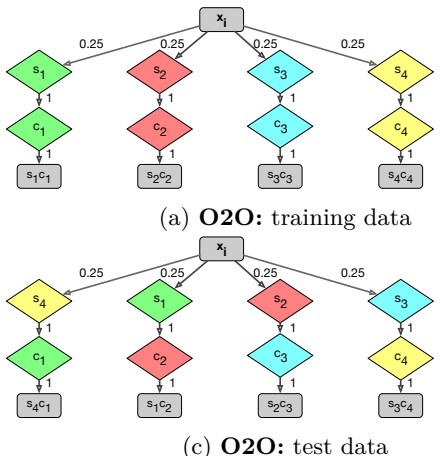

(a) **O2O:** training data

(c) **O2O:** test data

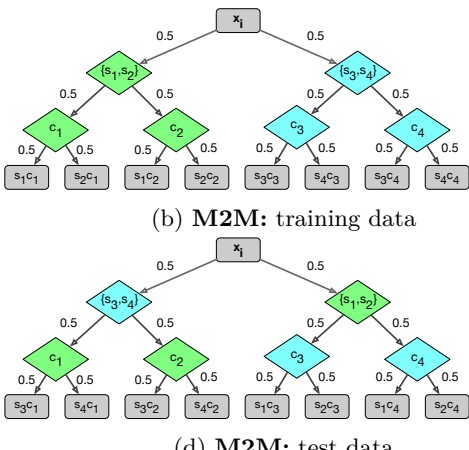

(b) **M2M:** training data

(d) **M2M:** test data

Figure 2: **Data distributions for our challenges:** $x_i$ is a random image sampled, each $s_i$ is a spurious attribute, and each $c_i$ is a class label. The edges indicate the probability that the sample $x_i$ has a given property, conditional on previous steps in the tree. The leaf nodes indicate the possible attribute-class combinations in the distribution. The colors emphasize the distribution shift in the test data.

methods on Spawrious in Section 5.1, with ERM being the worst and 10.68% points worse than the best method on average.

Like DomainBed, OoD-Bench (Ye et al., 2022) combines previously published datasets with the added contribution of characterizing them as a combination of diversity shift and style shift, allowing the evaluation of algorithms on a more comprehensive range of shifts. Methods that handle both shifts, like (Huang et al., 2022), will consistently beat ERM. By testing on unseen backgrounds-foreground combinations while having correlated backgrounds, we can address the two types of shifts they describe, while most datasets only address one

| Dataset | DG | O2O-SC | M2M-SC | Synthetic | Dataset Size |
|---|---|---|---|---|---|
| CelebA-Hair Color Liu et al. (2015) | ✗ | ✓ | ✗ | ✗ | 162770 |
| Waterbirds Sagawa et al. (2019a) | ✗ | ✓ | ✗ | ✓ | 4795 |
| CMNIST Arjovsky et al. (2019) | ✓ | ✓ | ✗ | ✓ | 60000 |
| DomainBed* Gulrajani & Lopez-Paz (2021) | ✓ | ✗ | ✗ | ✗ | - |
| WILDS Koh et al. (2021) | ✓ | ✗ | ✗ | ✗ | - |
| NICO Zhang et al. (2023) | ✓ | ✗ | ✗ | ✗ | 25000 |
| MetaShift Liang & Zou (2022) | ✓ | ✗ | ✗ | ✗ | 12868 |
| **Spawrious** | ✓ | ✓ | ✓ | ✓ | 152000 |

Table 1: **Differences between Spawrious and other benchmarks**, according to whether they pose a Domain Generalization (DG), One-To-One- and/or Many-To-Many Spurious Correlations challenge.

type of shift. WILDS (Koh et al., 2021), NICO (Zhang et al., 2023), FOCUS (Kattakinda & Feizi, 2022), MetaShift (Liang & Zou, 2022) collect in-the-wild data and group data points with environment labels. However, these benchmarks do not induce *explicit* spurious correlations between environments and labels. For example, WILDS-FMOW (Koh et al., 2021; Christie et al., 2017) possesses a label shift between non-African and African regions; yet, the test images pose a domain generalization (DG) challenge (test images were taken several years later than training images) instead of reverting the spurious correlations observed in the training data. Waterbirds (Sagawa et al., 2019a), and CelebA hair color (Liu et al., 2015; Izmailov et al., 2022) are binary classification datasets including spurious correlations but without unseen test domains (DG). Further, Idrissi et al. (2022) illustrates that a simple class-balancing strategy alleviates most of their difficulty, while Spawrious is class-balanced from the beginning. ColorMNIST (Arjovsky et al., 2019) includes spurious correlations and poses a DG problem. However, it is based on MNIST and, therefore, over-simplistic, i.e., it does not reflect real-world spurious correlations involving complex background features, such as the ones found in ImageNet (Singla & Feizi, 2022; Neuhaus et al., 2022). Hard ImageNet (Moayeri et al., 2022b) is a benchmark created by collecting images in ImageNet that contain spurious features, however, they do not satisfy our desiderata of multiple training environments and multiple difficulty levels Section 3. Like us, Li et al. (2023) create two synthetic datasets, UrbanCars and ImageNet-W, to test for spurious feature reliance, but these datasets do not satisfy our desiderata of photorealism and high-fidelity backgrounds Section 3. PUG (Bordes et al., 2023) synthetically generate a dataset of unfamiliar object-location images, but they do

not create a benchmark that introducese *explicit* spurious correlations between environment and labels. None of the above benchmarks include explicit training and test environments for M2M-SCs.

## 3 Benchmark Desiderata

Motivated by the shortcomings of previous benchmarks discussed in Section 2, we want first to posit some general desiderata that an improved benchmark dataset would satisfy. Next, we motivate and formalize the two types of spurious correlations we aim to study.

### 3.1 Six Desiderata

**1. Photo-realism**, unlike datasets containing cartoon/sketch images (Gulrajani & Lopez-Paz, 2021) or image corruptions (Hendrycks & Dietterich, 2019), which are known to conflict with current backbone network architectures (Geirhos et al., 2018a;b; Hermann et al., 2020), possibly confounding the evaluation of OOD algorithms. **2. Non-binary classification problem**, to minimize accidentally correct classifications achieved by chance. **3. Inter-class homogeneity and intra-class heterogeneity**, i.e., low variability *between* and high variability *within* classes, to minimize the margins of the decision boundaries inside the data manifold (Murphy, 2022). This desideratum ensures that the classification problem is non-trivial. **4. High-fidelity backgrounds** including complex features to reflect realistic conditions typically faced in the wild instead of monotone or entirely removed backgrounds (Xiao et al., 2020). **5. Access to multiple training environments**, i.e., the conditions of the *Domain Generalization* problem (Gulrajani & Lopez-Paz, 2021), which allow us to learn domain invariances, such that classifiers can perform well in novel test domains. **6. Multiple difficulty levels**, so future work can study cost trade-offs. For example, one may budget higher computational costs for methods succeeding on difficult datasets than those that succeed only on easy ones.

### 3.2 Spurious Correlations (One-To-One)

Here, we provide some intuition and discuss the conditions for a (one-to-one) spurious correlation (SC). We define a correlated, non-causal feature as a feature that frequently occurs with a class but does not cause the appearance of the class (nor vice versa). We abuse the term "correlated" as it is commonly used by previous work, but we consider non-linear relationships between two random variables too. Further, we call correlated features *spurious* if the classifier perceives them as a feature of the correlated class.

Next, we want to define a *challenge* that allows us to evaluate a classifier's harmful reliance on spurious features. Spurious features are not always harmful; even humans use context information to make decisions (Geirhos et al., 2020). However, a spurious feature becomes harmful if it alone is sufficient to trigger the prediction of a particular class without the class object being present in the image (Neuhaus et al., 2022).

To evaluate a classifier w.r.t. such harmful predictions, we evaluate its performance when the spurious correlations are reverted. The simplest setting is when a positive/negative correlation exists between one background variable and one label in the training/test environment.

---

**O2O-SC Challenge**

Let $p(\mathbf{X}, S, C)$ be a distribution over images $\mathbf{X} \in \mathbb{R}^D$, spurious attributes $S \in \mathcal{S} = \{s_1, \ldots, s_K\}$, and labels $C \in \mathcal{C} = \{c_1, \ldots, c_P\}$. Given $\hat{p}_{\text{data}} \neq p_{\text{test}}$, and $K = P$ it holds that for $i \in [K]$,

$$\text{corr}_{\hat{p}_{\text{data}}}\left(\mathbb{1}(S = s_i), \mathbb{1}(C = c_i)\right) > 0, \ \text{corr}_{p_{\text{test}}}\left(\mathbb{1}(S = s_i), \mathbb{1}(C = c_i)\right) < 0. \tag{1}$$

where the indicator function $\mathbb{1}(X = x)$ is non-zero when the *variable $X$* equals the *value $x$*.

---

Figure 1a illustrates the one-to-one (O2O) SC, in which pair-wise SCs between spurious features $S$ and labels $C$ exist within training environments, which then differ in the test environment.

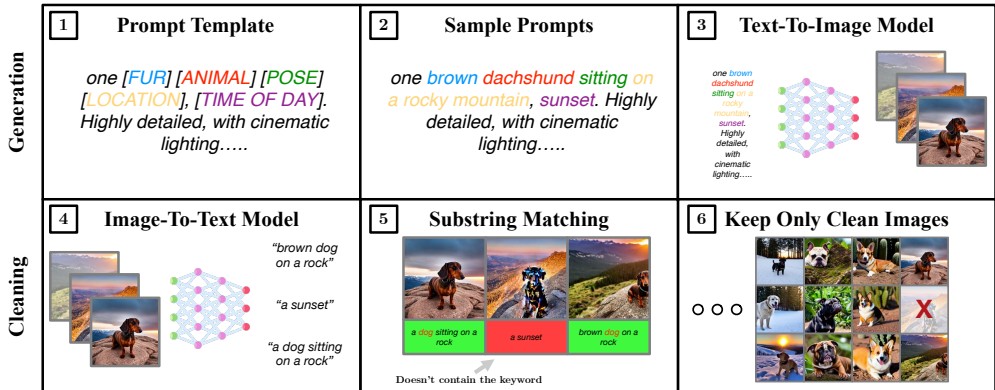

Figure 3: **Spawrious Pipeline:** We leverage text-to-image models for generation (Steps 1-3) and image-to-text models for cleaning of bad images (Steps 4-6). Details in Section 4.1 and Appendix E.

## 3.3 Many-To-Many Spurious Correlations

In this subsection, we conceptualize Many-To-Many (M2M) SCs, where the SCs hold over disjoint groups of spurious attributes and classes. For instance, in Figure 1b, each class from the class group {*Bulldog*, *Dachshund*} is observed with each background from the group {*Desert*, *Jungle*} in equal proportion in the training data.

Figure 2 shows an example of how to construct M2M-SCs, which contain richer spurious structures, following an *hierarchy* of the class groups correlating with spurious attribute groups. As we will see later in Section 4.3, the data-generating processes we instantiate for each challenge differ qualitatively.

---

**M2M-SC Challenge**

Consider $p(\mathbf{X}, S, C)$ defined in the O2O-SC Challenge. We further assume the existence of partitions $\mathcal{S} = \mathcal{S}_1 \dot{\cup} \mathcal{S}_2$ and $\mathcal{C} = \mathcal{C}_1 \dot{\cup} \mathcal{C}_2$. Given $\hat{p}_{\mathrm{data}}, p_{\mathrm{test}}$, it holds that for $j \in \{1, 2\}$

$$\mathrm{corr}_{\hat{p}_{\mathrm{data}}} \left( \mathbb{1}(S \in \mathcal{S}_j), \mathbb{1}(C \in \mathcal{C}_j) \right) = 1, \mathrm{corr}_{p_{\mathrm{test}}} \left( \mathbb{1}(S \in \mathcal{S}_j), \mathbb{1}(C \in \mathcal{C}_j) \right) = -1. \qquad (2)$$

---

# 4 The Spawrious Challenge

## 4.1 Dataset Construction

We instantiate the desiderata introduced in Section 3 by presenting *Spawrious*, a synthetic image classification dataset containing images of four dog breeds (classes) in six background locations (spurious attributes). Figure 3 summarizes the dataset construction pipeline, which we now discuss in more detail. The main idea is to leverage recently proposed text-to-image models (Rombach et al., 2022) for photo-realistic image generation and image-to-text models (NLP Connect, 2022) for filtering out low-quality images. We address potential ethical concerns that may arise from using a generative model to construct this dataset in Appendix A.

A **prompt template** allows us to define high-level factors of variation. We then **sample prompts** by filling in randomly sampled values for these high-level factors. The **text-to-image model** generates images given a sampled prompt; we use *Stable Diffusion v1.4* (Rombach et al., 2022). We pass the raw, generated images to an **image-to-text (I2T) model** to extract a concise description; here, we use the ViT-GPT2 image captioning model (NLP Connect, 2022). We perform a form of **substring matching** by checking whether important keywords are present in the caption, e.g., *"dog"*. This step avoids including images without class objects, which we sometimes observed due to the T2T model ignoring parts of the input prompt. We **keep only "clean" images** whose captions include important keywords. More details on this pipeline and possible

failures are discussed in Appendix E, as well as a measure of the accuracy of the prompt-image alignment in Appendix F.

## 4.2 Selecting train-test combinations

A priori, it is not apparent how the difficulty levels will vary across different combinations of training and test environments. To elucidate this matter, we conduct comprehensive evaluations over a range of combinations, utilizing a ResNet50 architecture trained with empirical risk minimization. Interestingly, we observe significant disparities in the difficulty levels of the combination splits. Notably, this trend in performance persisted irrespective of the training loss Table 3 employed. Hence, we present three difficulty levels for both O2O and M2M spurious correlations, with full details in Table 2. One hypothesis is that there exists a feature overlap in background features and core features that present difficulties to disentangle (Locatello et al., 2019).

## 4.3 Satisfying Benchmark Desiderata

To ensure **photorealism**, we generate images using *Stable Diffusion v1.4* (Rombach et al., 2022), trained on a large-scale real-world image dataset (Schuhmann et al., 2022), while carefully filtering out images without detectable class objects. We construct a 4-way classification problem to reduce the probability of accidentally correct classifications compared to a **binary classification problem** (e.g., CelebA hair color prediction or Waterbirds). Next, we chose dog breeds to reduce **inter-class variance**, inspired by the difference in classification difficulty between Imagenette (easily classified objects) (Howard, 2019a), and ImageWoof (Howard, 2019b) (dog breeds), two datasets based on subsets of ImageNet (Deng et al., 2009). We increase **intra-class variance** by adding animal poses to the prompt template.

We add "*[location] [time of day]*" variables to the prompt template to ensure **diverse backgrounds**, and select six combinations after careful experimentation with dozens of possible combinations, abandoning over-simplistic ones. Our final prompt template takes the form "*one [fur] [animal] [pose] [location], [time of day]. highly detailed, with cinematic lighting, 4k resolution, beautiful composition, hyperrealistic, trending, cinematic, masterpiece, close up*", and there are 72 possible combinations. The variables [location]/[animal] correspond to spurious backgrounds/labels for a specific background-class combination. The other variables take the following values: "*fur: black, brown, white, [empty]; pose: sitting, running, [empty]; time of day: pale sunrise, sunset, rainy day, foggy day, bright sunny day, bright sunny day*".

To construct **multiple training environments**, we randomly sample from a set of background-class combinations, which we further group by their **difficulty level into *easy, medium,* and *hard***. We construct two datasets for each SC type with $3,168$ images per background-class combination, thus 2 SC types $\times$ 4 environments $\times$ 6 difficulties $\times$ $3,168 = 152,064$ images in total.

**O2O-SC Challenge** We select combinations such that each class is observed with two backgrounds, spurious $b^{\mathrm{sp}}$ and generic $b^{\mathrm{ge}}$. For all images with class label $c_i$ in the training data, $\mu\%$ of them have the spurious background $b_i^{\mathrm{sp}}$ and $(100 - \mu)\%$ of them have the generic background $b^{\mathrm{ge}}$. Importantly, each spurious background is observed with only one class ($\hat{p}_{\mathrm{data}}(b_i^{\mathrm{sp}} \mid c_j) = 1$ if $i = j$ and 0 for $i \neq j$), while the generic background is observed for all classes with equal proportion. We train on two separate environments (with distinct data) that differ in their $\mu$ values. Thus, the change in this proportion should serve as a signal to a robustness-motivated optimization algorithm (e.g. IRM (Arjovsky et al., 2019), GroupDRO (Sagawa et al., 2019a) etc.) that the correlation is spurious.

For instance, in Figure 1a, training environment 1, 97% of the *Bulldog* images have spurious *Jungle* backgrounds, while 3% have generic *Beach* backgrounds. The spurious background changes depending on the class, but the relative proportions between each trio $c_i, b_i^{\mathrm{sp}}$ and $b_i^{\mathrm{ge}}$ are the same. In training env. 2, the proportions change to 87% and 13% split of spurious and generic backgrounds.

**M2M-SC Challenge** First, we construct disjoint background and class groups $\mathcal{S}_1, \mathcal{S}_2, \mathcal{C}_1, \mathcal{C}_2$, each with two elements. Then, we select background-class combinations for the training data such that for each class $c \in \mathcal{C}_i$, we pick a combination $(s, b)$ for each $s \in \mathcal{S}_i$. Second, we introduce two environments as shown in Figure 1b.

| Class | Train Env 1 | Train Env 2 | Test | Train Env 1 | Train Env 2 | Test | Train Env 1 | Train Env 2 | Test |
|---|---|---|---|---|---|---|---|---|---|
| | **O2O-Easy** | | | **O2O-Medium** | | | **O2O-Hard** | | |
| Bulldog | 97% De 3% B | 87% De 13% B | 100% Di | 97% M 3% De | 87% M 13% De | 100% J | 97% J 3% B | 87% J 13% B | 100% M |
| Dachshund | 97% J 3% B | 87% J 13% B | 100% S | 97% B 3% De | 87% B 13% De | 100% Di | 97% M 3% B | 87% M 13% B | 100% S |
| Labrador | 97% Di 3% B | 87% Di 13% B | 100% De | 97% Di 3% De | 87% Di 13% De | 100% B | 97% S 3% B | 87% S 13% B | 100% De |
| Corgi | 97% S 3% B | 87% S 13% B | 100% J | 97% J 3% De | 87% J 13% De | 100% S | 97% De 3% B | 87% De 13% B | 100% J |
| | **M2M-Easy** | | | **M2M-Medium** | | | **M2M-Hard** | | |
| Bulldog | 100% Di | 100% J | 50% S 50% B | 100% De | 100% M | 50% Di 50% J | 100% B | 100% S | 50% De 50% M |
| Dachshund | 100% J | 100% Di | 50% S 50% B | 100% M | 100% De | 50% Di 50% J | 100% B | 100% S | 50% De 50% M |
| Labrador | 100% S | 100% B | 50% Di 50% J | 100% Di | 100% J | 50% De 50% M | 100% M | 100% De | 50% B 50% S |
| Corgi | 100% B | 100% S | 50% Di 50% J | 100% J | 100% Di | 50% De 50% M | 100% M | 100% De | 50% B 50% S |

Table 2: **Proportions of Spurious Backgrounds By Class and Environment.** Backgrounds include: Beach (B), Desert (De), Dirt (Di), Jungle (J), Mountain (M), Snow (S).

| Method | One-To-One SC | | | Many-To-Many SC | | | Average |
|---|---|---|---|---|---|---|---|
| | **Easy** | **Medium** | **Hard** | **Easy** | **Medium** | **Hard** | |
| ERM (Vapnik, 1991) | $77.49\%_{\pm0.05}$ | $76.60\%_{\pm0.02}$ | $71.32\%_{\pm0.09}$ | $83.80\%_{\pm0.01}$ | $53.05\%_{\pm0.03}$ | $58.70\%_{\pm0.04}$ | 70.16% |
| GroupDRO (Sagawa et al., 2019a) | $80.58\%_{\pm0.74}$ | $75.96\%_{\pm2.18}$ | $76.99\%_{\pm2.60}$ | $79.96\%_{\pm2.79}$ | $61.01\%_{\pm4.64}$ | $60.86\%_{\pm1.71}$ | 72.56% |
| IRM (Arjovsky et al., 2019) | $75.45\%_{\pm2.57}$ | $76.39\%_{\pm2.22}$ | $74.90\%_{\pm1.27}$ | $76.15\%_{\pm2.83}$ | $67.82\%_{\pm4.39}$ | $60.93\%_{\pm1.09}$ | 71.94% |
| CORAL (Sun & Saenko, 2016) | $89.66\%_{\pm1.23}$ | $81.05\%_{\pm1.20}$ | $79.65\%_{\pm1.82}$ | $81.26\%_{\pm1.61}$ | $65.18\%_{\pm4.85}$ | $67.97\%_{\pm0.91}$ | 77.46% |
| CausIRL (Chevalley et al., 2022) | $89.32\%_{\pm1.20}$ | $78.64\%_{\pm0.62}$ | $80.40\%_{\pm1.32}$ | $85.76\%_{\pm1.02}$ | $63.15\%_{\pm2.98}$ | $68.93\%_{\pm0.28}$ | 77.20% |
| MMD-AAE (Li et al., 2018) | $78.81\%_{\pm0.02}$ | $75.33\%_{\pm0.03}$ | $72.66\%_{\pm0.01}$ | $80.55\%_{\pm0.02}$ | $59.43\%_{\pm0.04}$ | $54.39\%_{\pm0.05}$ | 70.20% |
| Fish (Shi et al., 2021) | $77.51\%_{\pm1.58}$ | $77.72\%_{\pm2.82}$ | $74.73\%_{\pm2.40}$ | $81.60\%_{\pm3.44}$ | $59.43\%_{\pm1.96}$ | $58.94\%_{\pm2.56}$ | 72.26% |
| VREx (Krueger et al., 2020) | $84.69\%_{\pm1.69}$ | $77.56\%_{\pm0.62}$ | $75.41\%_{\pm2.67}$ | $81.22\%_{\pm1.25}$ | $54.28\%_{\pm5.42}$ | $59.21\%_{\pm5.08}$ | 72.06% |
| W2D (Huang et al., 2022) | $81.94\%_{\pm1.03}$ | $76.74\%_{\pm0.70}$ | $76.84\%_{\pm1.32}$ | $80.80\%_{\pm2.24}$ | $62.82\%_{\pm2.23}$ | $61.89\%_{\pm2.71}$ | 73.50% |
| JTT (Zheran Liu et al., 2021) | $\mathbf{90.24\%}_{\pm3.09}$ | $\mathbf{87.28\%}_{\pm0.91}$ | $\mathbf{87.41\%}_{\pm0.99}$ | $79.23\%_{\pm1.83}$ | $60.56\%_{\pm5.55}$ | $57.58\%_{\pm3.86}$ | 77.05% |
| Mixup (Xu et al., 2019) // random shuffle | $88.48\%_{\pm0.74}$ | $82.75\%_{\pm3.12}$ | $75.75\%_{\pm1.16}$ | $\mathbf{89.61\%}_{\pm0.66}$ | $\mathbf{77.23\%}_{\pm0.97}$ | $71.21\%_{\pm2.33}$ | **80.84%** |
| Mixup // LISA (Yao et al., 2022) | $88.64\%_{\pm0.51}$ | $80.83\%_{\pm1.33}$ | $72.54\%_{\pm1.07}$ | $87.24\%_{\pm2.51}$ | $71.78\%_{\pm0.31}$ | $\mathbf{72.97\%}_{\pm4.23}$ | 79.00% |

Table 3: **Results for Spawrious-{O2O,M2M}-{Easy, Medium, Hard} using ImageNet-pretrained ResNet-50:** JTT (Zheran Liu et al., 2021) performs the best across the O2O challenges, while Mixup methods (Xu et al., 2019) perform best across M2M challenges and overall attain the highest average.

**Strength of the Spurious Correlation**   In the O2O case, the background features and core features are equally as predictive when the correlation is set to 1, while in the M2M case, the background features are less predictive than the core features. Thereby, we set the strength of the spurious correlation to be less than 1 in the O2O challenge (Section 3.2) while equal to 1 in the M2M challenge (Section 3.3). For example, desert background features would be equally as predictive as bulldog features in O2O-Easy (Table 2) without additional data from the beach background in both environments. We thus vary the extent of the correlation between the desert features and the class label in this challenge so that the training algorithms can learn to rely on the core features in the classification problem. In M2M-Easy (Table 2), bulldog features are much more predictive than dirt features when the M2M correlation is 1, with dirt features only present in half of the bulldog images. Then, we expect the model to rely more on the core features.

## 5   Experiments

We fine-tune a ResNet50 (He et al., 2016) model pre-trained on ImageNet, following previous work on domain generalization (Dou et al., 2019; Li et al., 2019; Gulrajani & Lopez-Paz, 2021). Given the size of our dataset, in preliminary experiments, we also tried training a ResNet50 from scratch, which consistently led to worse results. See Appendix B for analysis on the effect of ImageNet pretraining. We use various popular OOD methods, as listed below.

**Methods**   The field of worst-group-accuracy optimization is thriving with a plethora of proposed methods, making it impractical to compare all available methods. We choose the following six popular methods and their `DomainBed` implementation (Gulrajani & Lopez-Paz, 2021). **ERM** (Vapnik, 1991) refers to the canonical, average-accuracy-optimization procedure, where we treat all groups identically and ignore group labels, not targeting to improve the worst group performance. **CORAL** (Sun & Saenko, 2016) penalizes differences in the first and second moment of the feature distributions of each group. **IRM** (Arjovsky et al.,

2019) is a causality-inspired (Kaddour et al., 2022b) invariance-learning method, which penalizes feature distributions that have different optimal linear classifiers for each group. **CausIRL** (Chevalley et al., 2022) is another causally-motivated algorithm for learning invariances, whose penalty considers only one distance between mixtures of latent features coming from different domains. **GroupDRO** (Sagawa et al., 2019a) uses Group-Distributional Robust Optimization to explicitly minimize the worst group loss instead of the average loss. **MMD-AAE** (Li et al., 2018) penalizes distances between feature distributions of groups via the maximum mean discrepancy (MMD) and learning an adversarial auto-encoder (AAE). **JTT** (Zheran Liu et al., 2021) runs ERM for a certain number of epochs, stops, then runs classifications on all the training samples; then the misclassifications are up-weighted in the loss, and training continues. **W2D** (Huang et al., 2022) upweights datapoints in the loss that have either high *feature loss* or *sample loss* . **VREx** (Krueger et al., 2020) penalizes variance between the environment-specific training losses. **Fish** (Shi et al., 2021) rewards large inner products between environment-specific training gradients. **Mixup** (Xu et al., 2019) linearly interpolates between two images' pixel values, and has been implemented with random shuffle (randomly mix images across environments and labels) and **LISA** (Yao et al., 2022) (alternate between mixing across environments for the same label, or across labels for the same environment).

**Hyper-parameter tuning**   We follow the hyper-parameter tuning process used in `DomainBed` (Gulrajani & Lopez-Paz, 2021) with a minor modification. We keep the dropout rate (0.1) and the batch size fixed (128 for ResNets and 64 for ViTs) because we found them to have only a very marginal impact on the performance. We tune the learning rate and weight decay on ERM with a random search of 20 random trials. For all other methods, we further tune their method-specific hyper-parameters with a search of 10 random trials. We perform model selection based on the training domain validation accuracy of a subset of the training data. We reuse the hyper-parameters found for Spawrious-{O2O}-{Easy} and Spawrious-{M2M}-{Hard} on Spawrious-{O2O}-{Medium, Hard} and Spawrious-{M2M}-{Easy, Medium}, respectively. We also initially explored the ViT (Dosovitskiy et al., 2020) architecture, with results shown in Appendix C. Due to its poor performance, we chose to focus on ResNet50 results.

**Evaluation**   We evaluate the classifiers on a test environment where the SCs present during training change, as described in Table 2. For O2O, multiple ways exist to choose a test data combination; we evaluate one of them as selected using a random search process. In M2M, because there are only two class groups and two background groups, we only need to swap them as seen in Figure 1b.

## 5.1   Results

We find that JTT performs the best on the O2O challenges while being one of the worst methods on the M2M challenges. Within the M2M challenge, we find Mixup to perform the best, for both random shuffle and LISA, and overall Mixup attains the best average. This result contributes to the debate whether, for a fixed architecture, most robustness methods perform about the same (Gulrajani & Lopez-Paz, 2021) or not (Wiles et al., 2021). The performances of most methods get consistently worse as the challenge becomes harder. Most often, the data splits of our newly formalized M2M-SC are significantly more challenging than the O2O splits, most notably *M2M-{Hard, Medium}*. We conjecture that there is a strong need for new methods targeting such. {ERM, GroupDRO} and {CORAL, CausIRL} perform about the same, despite much different robustness regularization. All methods consistently achieve 98-99% in-distribution test performance (not shown in Table 1 to save space) despite differences in OOD performance. ERM performs worst on average for the ResNet50 set of results.

## 5.2   Misclassification analysis

In Section 5.1, we learned that ERM performs particularly poorly on both hard challenges. Now, we want to investigate why by examining some of the misclassifications. For example, we observe in Figure 4 that on the test set, the class *"Bulldog"* is misclassified as the classes whose most common training set background is the same as *"Bulldog"*'s test backgrounds.

Note that for all classes and in all data groups, both training and test environments, the number of data points per class is always balanced; rendering methods like *Subsampling large classes* (Idrissi et al., 2022),

**O2O-Hard:** Train Data: Corr(Dachshund, Mountains$) > 0$          **M2M-Hard:** Train Data: Corr(Labrador, Snow$) > 0$

**Misclassification: "*Dachshund*"**                              **Misclassification: "*Labrador*"**

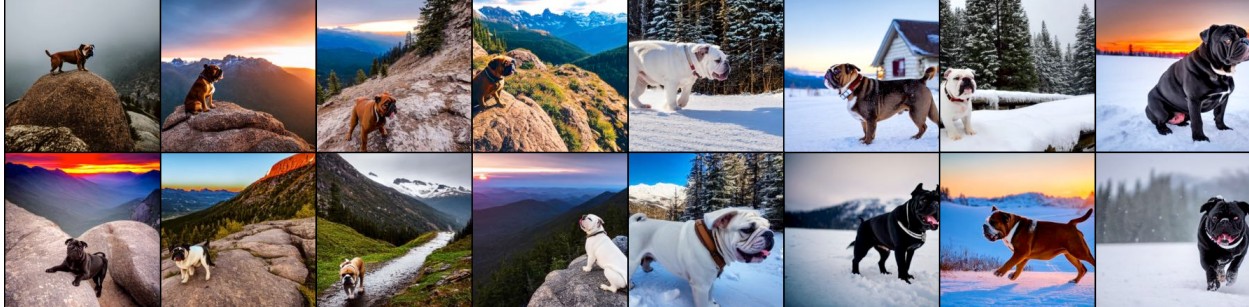

Figure 4: **ERM misclassifications due to spurious correlations.** The shown test images correspond to the class *"Bulldog"* with spurious backgrounds *"Mountains"* in the O2O-Hard (left) and *"Snow"* in the M2M-Hard (right) challenge.

which achieve state-of-the-art performance on other SC benchmarks, inapplicable. Hence, we conjecture that despite balanced classes, the model heavily relies on the spurious features of the *"Mountains"* and *"Snow"* backgrounds.

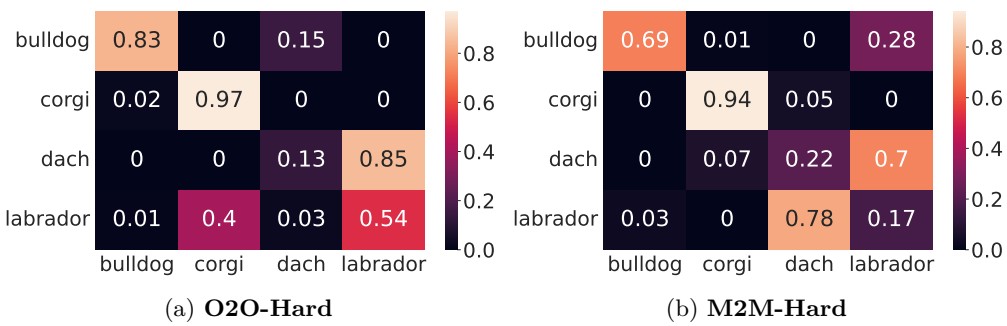

(a) **O2O-Hard**                              (b) **M2M-Hard**

Figure 5: **Confusion matrices for ERM models.** *X*-axis: predictions; *Y*-axis: true labels.

We further corroborate that claim by examining the model's confusion matrix in Figure 5. For example, Figure 5a shows the highest non-diagonal value for actual *"Dachshund"* images being wrongly classified as *"Labrador"*. We conjecture the reason being that in O2O-Hard, the background of *"Dachshund"* in the test set is *"Snow"*, which is the most common background of the training images of *"Labrador"*, as shown in Table 2. We examine the features learned by the ERM model using saliency maps in Appendix D.

## 6   Related Work

We summarized related benchmarks in Section 2. Further, we outline some works closest to ours here.

**Out-of-distribution Generalization**   approaches involve training a model simultaneously on multiple related but different domains, exploiting additional environment index labels in the training data (Ben-David et al., 2010; Blanchard et al., 2011; Muandet et al., 2013; Arjovsky et al., 2019), which our benchmark provides too. In order to design effective training losses, approaches may optimize the loss on the worst performing environment (Sagawa et al., 2019a), or enforce an invariance constraint, such as on the features (Sun & Saenko, 2016; Arjovsky et al., 2019; Chevalley et al., 2022) or on the gradients (Rame et al., 2022a). We discuss the methods we applied to our benchmark in Section 5.

**Spurious Correlations** have a long history in mathematical statistics (Pearson, 1897; Simon, 1954) and recently entered the machine learning discourse Sagawa et al. (2019b; 2020); Izmailov et al. (2022). They have been detected in common image classification settings via the usage of saliency maps (Moayeri et al., 2022a; Singla & Feizi, 2022). We use saliency maps to validate that an ERM model trained on Spawrious learned dependence on the spurious background feature in Appendix D.

**Causal Inference** The theory of causation provides another perspective on the sources and possible mitigations of spurious correlations (Peters et al., 2016; 2017; Kaddour et al., 2022b). Namely, we can formalize environment-specific data as samples from different interventional distributions, which keep the influence of variables not affected by the corresponding interventions invariant. This perspective has motivated several invariance-learning methods that make causal assumptions on the data-generating process (Arjovsky et al., 2019; Kaddour et al., 2022b). The field of treatment effect estimation also deals with mitigating spurious correlations from observational data (Chernozhukov et al., 2018; Künzel et al., 2019; Kaddour et al., 2021; Nie & Wager, 2021).

**Test-time domain adaptation with labels** involves either fine-tuning a model Rosenfeld et al. (2022); Izmailov et al. (2022); Kirichenko et al. (2023) or in-context learning Dong et al. (2022) to leverage a small amount of labeled test-domain examples.

**Miscellaneous** Nagarajan et al. (2020) analyze two different kinds of spurious correlations: *geometric* and *statistical* skew. Geometric skew occurs when there is an imbalance between groups of types of data points (i.e., data points from different environments) and leads to misclassification when the balance of groups changes. This understanding has motivated simply removing data points from the training data to balance between groups of data points (Arjovsky et al., 2022). In contrast, we study two particular types of SCs, which persist in degenerating generalization performance despite perfect balances of classes among groups. Further, Ye et al. (2022) provide a two-dimensional decomposition of OOD difficulty into correlation and diversity shifts between the training and test set. The challenges in our work span both of these dimensions, because the test environment contains unseen background-foreground combinations, a diversity shift, and the background is spuriously correlated with the foreground in the training data, a correlation shift.

## 7 Limitations and Future Work

The main limitations of our work have to do with how flexible the dataset can be. Spurious correlations can include **non-background** spurious attributes which currently are not covered. For example, Neuhaus et al. (2022) find that in the ImageNet (Deng et al., 2009) dataset, the class *"Hard Disc"* is spuriously correlated with *"label"*; however, *"label"* is not a background feature but rather part of the classification object. Spurious correlations also exist in **other data modalities**, e.g., text classification, leveraging the text generation capabilities of large language models (Brown et al., 2020). Other limitations of our work include evaluating **more generalization techniques** on Spawrious, including different robustness penalties (Liu et al., 2021; Blumberg et al., 2019; Krueger et al., 2021; Cha et al., 2021; Mahajan et al., 2021; Izmailov et al., 2022; Rame et al., 2022a), environment inference (Creager et al., 2021; Li et al., 2022; Sohoni et al., 2022; Huang et al., 2022), meta-learning (Zhang et al., 2020; Collins et al., 2020; Kaddour et al., 2020; Wang et al., 2021; Jiang et al., 2023), unsupervised domain adaptation (Ganin & Lempitsky, 2015; Long et al., 2016; Xu et al., 2021), dropout (LaBonte et al., 2022), flat minima (Cha et al., 2021; Kaddour et al., 2022a), weight averaging (Rame et al., 2022b; Wortsman et al., 2022; Kaddour, 2022), (counterfactual) data augmentation (Kaddour et al., 2022b; Gowal et al., 2021; Yao et al., 2022; Yin et al., 2023), fine-tuning of only specific layers (Kirichenko et al., 2022; Lee et al., 2023), diversity (Teney et al., 2022; Rame et al., 2022b), etc. Lastly, there is a possibility of **bias** creeping into the dataset via the generative model. Chuang et al. (2023) and others (Teo & Cheung, 2021; Zhao et al., 2018) have studied debiasing techniques for vision-language models, such as *Stable Diffusion v1*, and have moderate success in removing unexpected sources of spurious correlations.

## 8 Conclusion

We present Spawrious, an image classification benchmark with two types of spurious correlations, one-to-one (O2O) and many-to-many (M2M). We carefully design six dataset desiderata and instantiate them by leveraging recent advances in text-to-image and image captioning models. Next, we conduct experiments, and our findings indicate that even state-of-the-art group robustness techniques are insufficient in handling Spawrious, particularly in scenarios with Hard-splits where accuracy is below 73%. Our analysis of model errors revealed a dependence on irrelevant backgrounds, thus underscoring the difficulty of our dataset and highlighting the need for further investigations in this area. A more extensive discussion of limitations and future work can be found in Section 7.

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

# A   Ethical Concerns

## A.1   Biases

We first acknowledged that generative models can inherit biases from their training data, including those related to dog breed representation and dog breed characteristics. We utilized various measures to mitigate these biases:

- *Dog Breed Representation:* By design, we ensured that the breeds in our dataset are balanced, avoiding underrepresentation or overrepresentation of any particular breed.

- *Dog Breed Characteristics:* We examined the characteristics associated with each breed and verified that our model does not exaggerate or stereotype them.

Further, we employed quality control measures, as described in Section 4.1, to guarantee that images are realistic and high-quality, regardless of breed. We manually reviewed the generated images to ensure they were free from harmful associations and stereotypes.

## A.2   Copyright Considerations

We purposefully decided to use StableDiffusion, which offers a permissive license that allows for commercial and non-commercial usage. See more info in (Rombach & Esser, 2022).

Further, we are aware of possible copyright and fair use offenses, which are still debated. To our knowledge, under US law, fair uses of in-copyright works do not infringe copyrights Samuelson (2023). Courts consider four factors when assessing fair use defenses: (1) the purpose of the challenged use, (2) the nature of the copyrighted works, (3) the amount and substantiality of the taking, and (4) the effect of the challenged use on the market for or value of the copyrighted work, which we address as follows:

1. *Purpose and character*: Academic research is nonprofit and educational.

2. *Nature of the work*: Academic research often involves factual or informational works.

3. *Amount and substantiality*: We use generated images, which are likely to include only small portions if any of copyrighted works (Carlini et al., 2023; Somepalli et al., 2023).

4. *Effect on the market*: Academic research is unlikely to harm the market for the original work.

# B   Effect of ImageNet Pre-Training

| Method | One-To-One SC | | | Many-To-Many SC | | | Average |
|---|---|---|---|---|---|---|---|
| | Easy | Medium | Hard | Easy | Medium | Hard | |
| ERM | $45.75\%_{\pm1.26}$ | $46.86\%_{\pm1.10}$ | $\mathbf{41.85\%_{\pm0.56}}$ | $57.67\%_{\pm2.55}$ | $30.03\%_{\pm0.28}$ | $30.05\%_{\pm1.34}$ | $42.04\%$ |
| GroupDRO | $\mathbf{46.50\%_{\pm0.91}}$ | $46.52\%_{\pm0.95}$ | $39.80\%_{\pm1.66}$ | $60.82\%_{\pm0.58}$ | $31.72\%_{\pm0.35}$ | $\mathbf{31.62\%_{\pm1.72}}$ | $\mathbf{42.83\%}$ |
| MMD-AAE | $44.09\%_{\pm1.80}$ | $\mathbf{46.87\%_{\pm1.46}}$ | $39.67\%_{\pm0.84}$ | $\mathbf{61.24\%_{\pm0.93}}$ | $\mathbf{32.10\%_{\pm0.47}}$ | $30.77\%_{\pm1.58}$ | $42.46\%$ |
| ERM | $77.49\%_{\pm0.05}$ | $\mathbf{76.60\%_{\pm0.02}}$ | $71.32\%_{\pm0.09}$ | $\mathbf{83.80\%_{\pm0.01}}$ | $53.05\%_{\pm0.03}$ | $58.70\%_{\pm0.04}$ | $70.16\%$ |
| GroupDRO | $\mathbf{80.58\%_{\pm0.74}}$ | $75.96\%_{\pm2.18}$ | $\mathbf{76.99\%_{\pm2.60}}$ | $79.96\%_{\pm2.79}$ | $\mathbf{61.01\%_{\pm4.64}}$ | $\mathbf{60.86\%_{\pm1.71}}$ | $\mathbf{72.56\%}$ |
| MMD-AAE | $78.81\%_{\pm0.02}$ | $75.33\%_{\pm0.03}$ | $72.66\%_{\pm0.01}$ | $80.55\%_{\pm0.02}$ | $59.43\%_{\pm0.04}$ | $54.39\%_{\pm0.05}$ | $70.20\%$ |

Table 4: **Impact of ImageNet pretraining:** ResNet-50 without ImageNet pretraining (top) vs ResNet-50 with ImageNet pretraining (bottom) results

We have included ImageNet pretraining for all of our main body results in Table 1, as has been done for results comparisons on Waterbirds (Sagawa et al., 2019a) and CelebA (Liu et al., 2015) and has become standard

practice for image classification (Krizhevsky et al., 2012). However, we also measure the performance of a ResNet50 trained just on the Spawrious challenges and report our results in Table 4. We find that pretraining makes a consistently positive impact on the performance of the classifiers, with a 28.12% point difference between the ERM performances.

## C  Effect of Model Architecture

| Method | One-To-One SC | | | Many-To-Many SC | | | Average |
|---|---|---|---|---|---|---|---|
| | **Easy** | **Medium** | **Hard** | **Easy** | **Medium** | **Hard** | |
| ERM | $36.28\%_{\pm 1.17}$ | $32.78\%_{\pm 2.55}$ | $30.2\%_{\pm 0.83}$ | $55.56\%_{\pm 0.75}$ | $\mathbf{32.78\%_{\pm 2.55}}$ | $\mathbf{30.20\%_{\pm 0.83}}$ | 40.44% |
| GroupDRO | $\mathbf{41.14\%_{\pm 1.62}}$ | $\mathbf{51.43\%_{\pm 0.53}}$ | $\mathbf{40.21\%_{\pm 1.76}}$ | $53.79\%_{\pm 1.35}$ | $30.79\%_{\pm 1.75}$ | $25.45\%_{\pm 1.15}$ | 40.47% |
| MMD-AAE | $40.64\%_{\pm 3.11}$ | $53.36\%_{\pm 0.95}$ | $38.54\%_{\pm 1.92}$ | $\mathbf{58.42\%_{\pm 1.77}}$ | $24.75\%_{\pm 0.59}$ | $28.91\%_{\pm 2.68}$ | **40.77%** |
| ERM | $77.49\%_{\pm 0.05}$ | $\mathbf{76.60\%_{\pm 0.02}}$ | $71.32\%_{\pm 0.09}$ | $\mathbf{83.80\%_{\pm 0.01}}$ | $53.05\%_{\pm 0.03}$ | $58.70\%_{\pm 0.04}$ | 70.16% |
| GroupDRO | $\mathbf{80.58\%_{\pm 0.74}}$ | $75.96\%_{\pm 2.18}$ | $\mathbf{76.99\%_{\pm 2.60}}$ | $79.96\%_{\pm 2.79}$ | $\mathbf{61.01\%_{\pm 4.64}}$ | $\mathbf{60.86\%_{\pm 1.71}}$ | **72.56%** |
| MMD-AAE | $78.81\%_{\pm 0.02}$ | $75.33\%_{\pm 0.03}$ | $72.66\%_{\pm 0.01}$ | $80.55\%_{\pm 0.02}$ | $59.43\%_{\pm 0.04}$ | $54.39\%_{\pm 0.05}$ | 70.20% |

Table 5: **Impact of Vit-B instead of ResNet-50:** Vit-B pretrained on ImageNet (top) vs ResNet-50 pretrained on ImageNet (bottom) results

We experiment with the ViT-B/16 (Dosovitskiy et al., 2020), following (Izmailov et al., 2022; Mehta et al., 2022). Based on Table 5, we make the following observations: The ViT backbone architecture worsens the performance for both MMD-AAE and ERM, underperforming the ResNet50. The best results for ERM were obtained with ResNet50, which performs 29.72% points better than the best ViT. In the debate on whether ViTs (Dosovitskiy et al., 2020) are generally more robust to SCs (Ghosal et al., 2022) than CNNs or not (Izmailov et al., 2022; Mehta et al., 2022), our results side with the latter. We observe that a ViT-B/16 pretrained on ImageNet22k had worse test accuracies than the ResNet architecture.

## D  Saliency maps for misclassifications

Saliency maps (Simonyan et al., 2013; Zhou et al., 2015; Selvaraju et al., 2019; Omeiza et al., 2019) are a method for investigating the input features that most positively affect a model's particular classification. We applied the Smooth Grad-CAM++ saliency map method (Omeiza et al., 2019; Fernandez, 2020) to the misclassified images from an ERM model in the test domains of the O2O-Hard and M2M-Hard challenges. The saliency maps we obtained in Figure 6 and Figure 7 suggest that the ERM model was sensitive to (spurious) background features, although seemingly more in the O2O challenge than the M2M challenge.

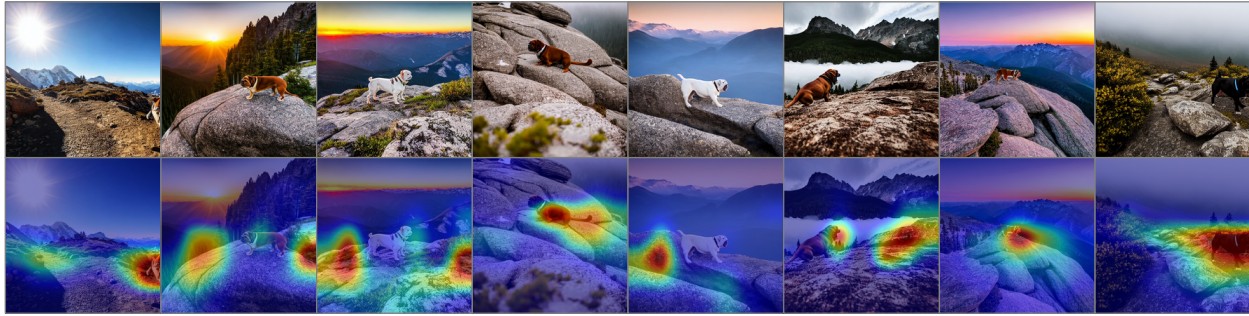

Figure 6: **O2O-Hard saliency maps:** all images were misclassifications of *Bulldog* as *Dachshund*

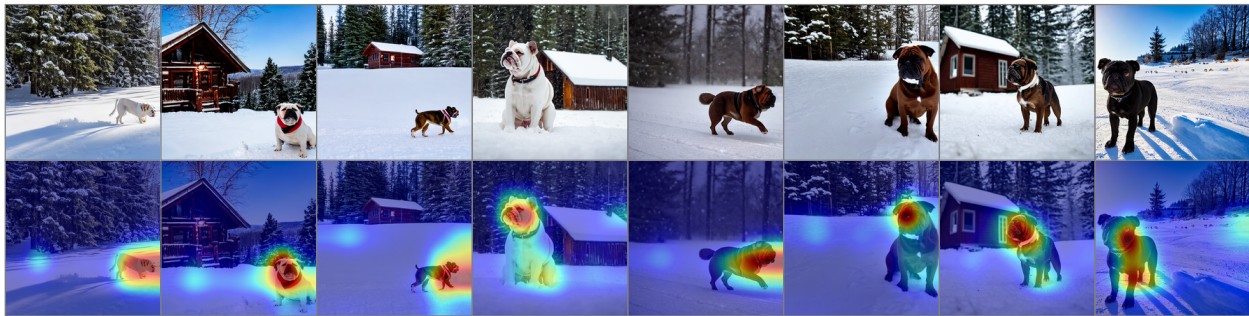

Figure 7: **M2M-Hard saliency maps:** all images were misclassifications of *Bulldog* as *Labrador*

Next, we compare qualitatively the difference in saliency maps between the Mixup and ERM optimization methods, which can be seen in Figure 8. While the exact saliency patter differs between the two methods, they ultimately seem to be attending to the same image features.

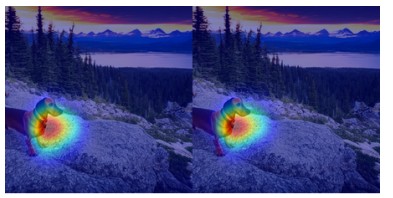
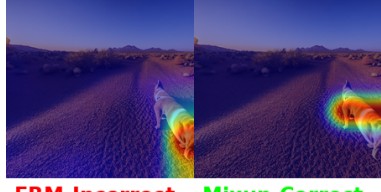
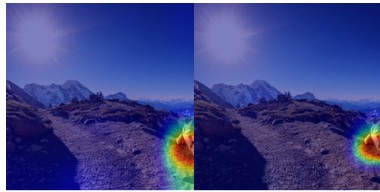

Figure 8: **Saliency comparisons between Mixup and ERM**

# E    Failure Analysis of the Generation Pipeline

We conduct a failure analysis in two ways: manual and automatic. In our manual visual examination, we inspected large samples of the generated images via human annotators (the authors). Our automated failure analysis pipeline is described in Section 4.2. For example, to test the quality of a prompt, we only accept it under two conditions: at least 95 images out of 100 look realistic and fit the prompt. Second, all remaining images must only be unfit because of the absence of a dog in the image. Identifying a dog in an image is a relatively easy task for the image captioning model. We confirmed by evaluating on the unfit images and assessing that they all get flagged by the image captioning model (the caption does not contain the word dog).

# F    Cleanliness Analysis of the Dataset

We have checked the accuracy of prompt-image alignment of images such as those in Figure 9 from a random sample of our dataset using human annotators (10 volunteers). We collected a random sample of 480 images from our dataset, appended with the intended caption for the image, and then partitioned this dataset into 10 folders. We asked 10 volunteers to scan the images and return a score for the number of correctly aligned images. Our scores were: 48, 46, 46, 46, 47, 47, 46, 46, 47, 48; resulting in an average of $46.7/48 = \mathbf{97.2}\%$.

# G    Discussion of M2M vs O2O

In order to understand how the M2M challenge leads to poor generalisation performance, consider the following situation, where the classifier achieves low loss in training by simulating a decision tree within the network, as depicted in Figure 2b of the submission. The model first represents the background, and then decides which group of dogs the image could be representing conditioned on the background. Within this

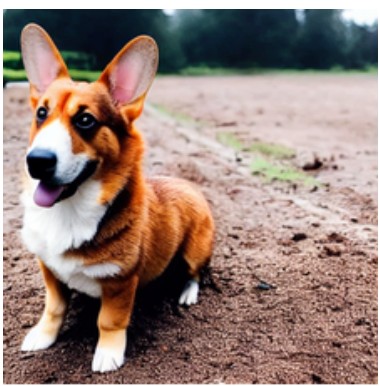 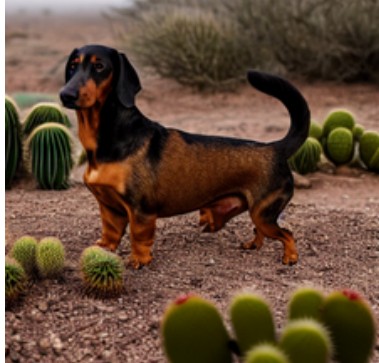 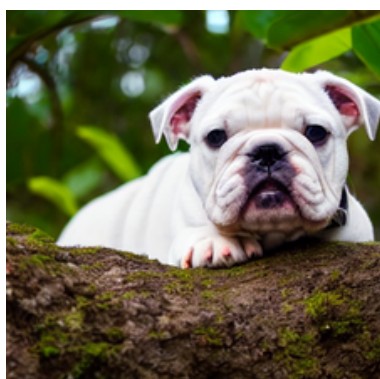

a corgi in the dirt location   a dachshund in the desert location   a bulldog in the jungle location

Figure 9: **Volunteers decided on prompt-image alignment for 224x224 images**: We asked 10 volunteers to scan images such as the three shown above and return a score for the number of correctly aligned images

setting, the spurious feature dependence arises at the beginning of the decision tree. In the test data, this decision tree fails to work because the background group is wholly unpredictive of the class groups. As seen in Figure 2d, the blue background group (s3, s4) is a feature used by the model to decide between classes (c3, c4), when in fact the model should be deciding between (c1, c2).

