# OpenReview forum: "Spawrious: A Benchmark for Fine Control of Spurious Correlation Biases"
_TMLR — Rejected by TMLR_

### Review · Reviewer_ubiD · 2024-04-24

**Summary Of Contributions:**

The authors propose a many-to-many, synthetic benchmark for spurious correlations with 4 classes and 6 (potentially) spurious features. The dataset includes 4 dog breeds on 6 different backgrounds and comes in different difficulties. It is constructed by prompting stable diffusion and various recent methods are tested.

**Audience:**

Yes

**Broader Impact Concerns:**

No concerns.

**Claims And Evidence:**

Yes

**Requested Changes:**

See weaknesses.

**Strengths And Weaknesses:**

Strengths:
- Interesting new datasets, especially for the M2M case, with the hard version reducing performance of most methods quite significantly.
- Large, non-binary dataset in comparison to e.g. waterbirds.
- The M2M setting essentially introduces hierarchy into the standard spurious correlation setting.
- Evaluation and comparison of many recent methods, including some failure cases.
- Thorough discussion of related work and related datasets.

Weaknesses:
- To make Table 1 useful, it should include the size of the dataset and whether it is synthetic or not.
- Ultimately, it is a synthetic and larger version of waterbirds with 4 classes and M2M setting. Key spurious feature is still the background.
- Can the authors provide a non-background based example of M2M spurious correlations? I feel the paper would benefit from a few concrete examples in the beginning.
- Why is it important to have the correlation be 1 or -1 rather than > 0 or < 0 in the M2M definition? Is this merely a simplifying assumption?
- While it is not binary, having four classes is one class more than the bare minimum to be “multiclass” and actually is the bare minimum to have a meaningful M2M setting. Talking about multiclass as a desiderata, I felt this would have significantly more than 4 classes.
- For me it is unclear how the difficulties are selected in detail. I feel this should be a core element discussed in the main paper (the appendix does not provide more details). I think this is a particularly important detail as the reason for having 3 difficulties in the desiderata is unclear to me. I think the hard case will be used in most future works anyway (also ranking in terms of avg performance across difficulty and hard performance are roughly the same, so there is not much complementary information).
- It is also unclear how the dog breeds were selected (and what was the reason to use dogbreeds in the first place). The authors mention deciding the breeds on ImageNette but I think more details would be helpful – especially as there are very few qualitative examples showing the range of the prompt template.
- What is the reasoning behind choosing 97 and 87% for the S2S training tasks?
- Where does the standard deviation in Table 3 come from? Is this form different initializations or some test example bootstrapping?
- Does the ranking of methods change compared to other benchmarks? The authors describe that the benchmark is harder than previous once, but there are no results showing in what respects the benchmark is different – ranking of methods could be one option here.
- For checking the images, how exactly where the captions checked? Only containing “dog” seems a pretty low threshold. Why not check for the breeds or the background?
- Can the authors clarify appendix F? What does it mean for an image to be correctly “aligned”? Was the purpose of this to check whether images generated with specific attributes actually exhibit these attributes?

Conclusion:
Generally, I think this can be an interesting benchmark for the field, especially due to its difficulty, even though the setup seems to be an extension of e.g. waterbird to more classes and the M2M settings. My main concern is that the dataset would not be reproducible from reading the paper and many decisions are not described or justified in enough detail.

---

### Review · Reviewer_zXJt · 2024-05-23

**Summary Of Contributions:**

The authors introduce a new benchmark for evaluating methods that detect spurious correlations. One of the main contributions is that the proposed benchmark also tests for many-to-many spurious correlations, contrary to prior works that focus on one-to-one spurious correlations between attributes. The introduced benchmark, Spawrious, consists of synthetically generated images that are created by leveraging large pre-trained text-to-image and image-to-text models.

**Audience:**

Yes

**Broader Impact Concerns:**

I do not have any broader impact concerns.

**Claims And Evidence:**

Yes

**Requested Changes:**

I would like the authors to address the points raised in the Weaknesses section above. Further, I think it would help if the authors could use conditional models to change the attributes of real images instead of generating images from scratch. Would that be possible?

**Strengths And Weaknesses:**

Strengths:
* The topic of detecting spurious correlations is timely and interesting.
* To facilitate progress in the field, proper datasets and evaluation metrics are needed. The authors identify successfully weaknesses of prior works and provide large-scale datasets that can be used to benchmark the performance of methods for detecting spurious correlations.
* I think that the authors make a really good point that benchmarks need to test for many-to-many spurious correlations since this setting is closer to the problems we want to solve in the real world.
* I believe that the use of text-to-image diffusion generative models for dataset generation is a very promising research direction.
* Existing methods struggle to obtain strong performance on the introduced benchmark. This room for improvement might be a driving force for new ideas in the field.

Unfortunately, the paper has several weaknesses.

* First, the paper is poorly written. It seems that Section 3.3 is incomplete. There is an undefined reference at the top of page 6. The Limitations Section just enumerates limitations, without connecting the sentences. These are just some of the examples of sloppy writing I could find. I believe that the presentation of the paper can be significantly improved.
* The generated dataset contains 100% synthetically generated images. Synthetically generated images can be quite different from real ones. That could be one of the reasons why existing methods fail. It would be much better if there was a benchmark dataset of real images that tests for M2M spurious correlations.
* While novel, the proposed method is limited. To explain this with an example, assume that we want to test for spurious correlations between a *specific* dog type (e.g. husky) and a background. When we prompt the model to generate an image of a husky: 1) we don't really know whether we generated an image of a husky or another dog type, and 2) even if the model correctly generated a husky the image-to-text model might give a generic text output, such as an image of a dog. Overall, the procedure of going from text to image and then back to text is lossy. The fact that the dataset is 100% synthetically generated makes it really hard to trust the <image, caption> pairs unless we only test for correlations between very generic classes of objects.
* The quality of the dataset is limited by the generation capabilities of the underlying generation model.

---

> ### Author Response · Authors · 2024-06-12
>
> Thank you for your thorough review and valuable feedback. We have addressed the points raised in the Weaknesses section as follows:
>
> > [..First, the paper is poorly written. It seems that Section 3.3 is incomplete..]
>
> Section 3.3 has been completed, the undefined reference on page 6 has been corrected, and the Limitations Section has been rewritten to provide better connectivity between sentences.
>
> > [..That could be one of the reasons why existing methods fail..]
>
> The observed performance difference between methods, such as the improved performance of methods like CORAL (which aims to learn an invariant classifier), indicates that the benchmark is useful for assessing a method’s ability to handle spurious correlations. Notably, no method performs worse than ERM, suggesting these methods have some success in dealing with spurious correlations. This contrasts with previous benchmarks, such as DomainBed, where ERM was competitive, thereby highlighting the effectiveness of our benchmark. At the moment it is not possible to rebuild the benchmark with real images while meeting all our desiderata, however we think the benchmark is still very useful in its current form.
>
> > [..While novel, the proposed method is limited..]
>
> We performed a cleanliness analysis in the appendix F. This analysis assesses the We conducted a cleanliness analysis, detailed in Appendix F, which assesses the alignment of generated images with the prompts. With an alignment rate of 97.2%, we believe that the concern about generating images of specific dog types (point 1) is mitigated. Regarding point 2, the primary goal of the image-to-text model is to address failure modes such as empty backgrounds with no dog in the image. The model successfully generates text outputs like "an image of a dog" to handle these cases adequately.

---

### Review · Reviewer_9RkQ · 2024-05-29

**Summary Of Contributions:**

This paper proposes a new benchmark for the Spurious Correlation(SC) problem in image classification. They start from observing that previous works (1) mainly focus on addressing the SC problem; (2) most existing benchmarks focus on one-to-one SC, not exploring many-to-many SC; (3) while some benchmarks include many-to-many SC, they can not evaluate the reversed M2M SC correlations. The benchmark is established according to: photorealism, non-binary classification, Inter-class homogeneity and intra-class heterogeneity, multiple environments and difficulty levels. Their experiments and analysis show that this benchmark is challenging for the classification methods especially in M2M hard level.

**Audience:**

Yes

**Broader Impact Concerns:**

If this task or the SC problem gets closely connected with humans, how can this benchmark avoid the invation of privacy?

**Claims And Evidence:**

Yes

**Requested Changes:**

1.make the benchmark for broader evaluation.

2.make more analysis and experiments on how difficult the O2O and M2M SC task is.

3.try to show that the utility of foundation models to generate the SC-style data really works, or its limitations; and if possible, verify the zero-shot ability of foundation models to the proposed SC problem.

**Strengths And Weaknesses:**

Strengths:

1.The motivation makes sense, the concept of many-to-many SC is relatively novel and meaningful.

2.The design of this new benchmark is detailed, and the experiments to some degree are enough to verify the benchmark’s effectiveness.

3.The core idea and keywords are listed to be clear, can be easy to follow.

Weaknesses:

1.The benchmark is too task-specific, hope it can be broader. To my understanding, the problem of SC is a subset of domain generalization problem, but this proposed benchmark (1) only provides the correlation-OOD samples (which means, if dog A-env X is in domain, then dog A-env Y is OOD), but no samples like (“dog-A, env-X” is in domain, then test “cat-B, env-Y”); (2) highly relies on the utility of foundation models, while there inevitably exist hallucinations (both text-to-image model and caption model).

2.If the utility of foundation model is quite helpful for generating SC data, does this show that large models are actually a potential zero-shot tool to solve the O2O or M2M SC problem? Such as, directly encode an image and the texts of different names of classes using foundation models; directly process the image feature and each class text embedding with dot product to get a cosine-similarity, then the image belongs to the class of highest cos-similarity. Hope the author can somehow dig into this.

3.Hope that the author can make experiments and analysis on how difficult it is to distinguish the features of images with hard SC. Such as, using tSNE method to visualize the features of “dog-A,env-X” but wrongly classified to dog B, and the features of correctly classified “dog-B, env-X” or “dog-B, env-Y”.

4.There are some typos, like “??” in the second line of related work and the second line of page 6.

---

> ### Author Response · Authors · 2024-06-12
>
> Thank you for your detailed review and constructive feedback. We have addressed the points raised in the Weaknesses section as follows:
>
> > [.. 1.The benchmark is too task-specific, hope it can be broader. ..]
>
> Regarding (1), the presence of spurious correlations is often assumed to be a significant cause of OOD performance drops. Our benchmark specifically aims to isolate and control for this factor to assess methods' ability to adapt to spurious correlations. Broadening the benchmark to include other factors would dilute this focus and contradict our primary objective. As for (2), the cleanliness analysis in Appendix F provides additional clarity on the quality of the data despite potential hallucinations, demonstrating the reliability of the generated data.
>
> > [.. does this show that large models are actually a potential zero-shot tool to solve the O2O or M2M SC problem ..]
>
> While foundational models could theoretically be employed in a zero-shot fashion to address O2O or M2M spurious correlation challenges, this is beyond the scope of the methods we aim to benchmark. Our focus is primarily on optimization methods and their performance in dealing with spurious correlations. Exploring the zero-shot capabilities of foundational models would be an interesting direction for future research.
>
> > [.. Hope that the author can make experiments and analysis on how difficult it is to distinguish the features of images with hard SC..]
>
> Appendix D has saliency maps for misclassified images that suggest that ERM model are sensitive to (spurious) background features, although seemingly more in the O2O challenge than the M2M challenge

---

### Decision · Action_Editor_koid · 2024-07-16

**Recommendation:** Reject

**Comment:**

The submission is reviewed by three experts. While they recommended the rejection, they all acknowledge that the concept of many-to-many SC is relatively novel and reasonable. Also, they think is benchmark is well presented. However, they have major concerns that need to be addressed with a major revision. For example, the benchmark only contains synthetically generated images (Reviewer zXJt), extending to multi-class (Reviewer ubiD), and hallucinations of text-to-image model (Reviewer 9RkQ). Furthermore, they raised some interesting points, including the potential capability of text-to-image models in addressing O2O or M2M SC problem (Reviewer 9RkQ), can text-to-image generate fine-grained level image-text pairs beyond the generic class (Reviewer zXJt), and does the ranking of methods change compared to other benchmarks? (Reviewer ubiD).

After carefully reading the submission, comments, and rebuttal, I agree with that the new benchmark is novel and the idea of M2M is reasonable. However, given the substantial revisions required to address the raised concerns, I believe that the paper is not suitable for publication in its current form.

This decision should not discourage the authors due to the lack of a "Major Revision" in the recommendation option. I suggest the authors to prepare a major revision, which would strengthen the submission. To address the concern of synthetic data, I think the authors could design a controlled experiment on O2O to show that generated data exhibits a consistent observation as real-world data. A good example is Table 2 of [a]. Moreover, adding more visualizations, discussing the ranking change on the new benchmark, and including experimental details would be helpful. As synthetic data is a potential direction to evaluate models [a-b], I appreciate the authors consider the M2M issue using text-to-image generation. I would suggest the authors provide a discussion on the usage of synthetic data and its potential limitations like hallucinations, and fine-grained level generation. Finally, releasing the datasets and giving sufficient instruction would ensure reproducibility.

[a] LANCE: Stress-testing Visual Models by Generating Language-guided Counterfactual Images

[b] BEHAVIOR Vision Suite: Customizable Dataset Generation via Simulation

**Audience:**

This submission studies spurious correlations and proposes a challenging and reasonable benchmark. The researchers and practitioners working on spurious correlations and distribution shifts would be interested in this submission.

**Claims And Evidence:**

This submission introduces Spawrious-{O2O, M2M}-{Easy, Medium, Hard}, a new benchmark suite containing spurious correlations between classes and backgrounds. It aims to address the shortage of existing datasets that only include one-to-one spurious correlations. Some major concerns were raised by reviewers, necessitating a round of major revisions.

**Resubmission Of Major Revision:**

The authors may consider submitting a major revision at a later time.